# Glycans are not necessary to maintain the pathobiological features of bovine spongiform encephalopathy

Alicia Otero[1][☯], Tomás Barrio[2][☯], Hasier Eraña[3,4], Jorge M. Charco[3,4], Marina Betancor[1], Carlos M. Díaz-Domínguez[3], Belén Marín[1], Olivier Andréoletti[2], Juan M. Torres[5], Qingzhong Kong[6], Juan J. Badiola[1], Rosa Bolea[1]*, Joaquín Castilla[3,7,8]*

**1** Centro de Encefalopatías y Enfermedades Transmisibles Emergentes, Universidad de Zaragoza, IA2, ISS Aragón, Zaragoza, Spain, **2** UMR INRAE-ENVT 1225 Interactions Hôtes-Agents Pathogènes (IHAP), Institute Nationale de Recherche pour l'Alimentation, l'Agriculture et l'Environnement (INRAE)—École Nationale Vétérinaire de Toulouse (ENVT), Université de Toulouse, Toulouse, France, **3** Center for Cooperative Research in Biosciences (CIC BioGUNE), Basque Research and Technology Alliance (BRTA), Prion Research Lab, Derio, Spain, **4** Atlas Molecular Pharma S. L., Derio, Spain, **5** Centro de Investigación en Sanidad Animal, CISA-INIA, Valdeolmos, Madrid, Spain, **6** Departments of Pathology and Neurology & National Center for Regenerative Medicine, Case Western Reserve University, Cleveland, Ohio, United States of America, **7** IKERBASQUE, Basque Foundation for Science, Bilbao, Spain, **8** Centro de Investigación Biomédica en Red de Enfermedades Infecciosas (CIBERINFEC), Carlos III National Health Institute, Madrid, Spain

☯ These authors contributed equally to this work.
* jcastilla@cicbiogune.es (JC); rbolea@unizar.es (RB).

**Data Availability Statement:** All relevant data are within the manuscript and its Supporting Information files.

## Abstract

The role of the glycosylation status of PrP$^C$ in the conversion to its pathological counterpart and on cross-species transmission of prion strains has been widely discussed. Here, we assessed the effect on strain characteristics of bovine spongiform encephalopathy (BSE) isolates with different transmission histories upon propagation on a model expressing a non-glycosylated human PrP$^C$. Bovine, ovine and porcine-passaged BSE, and variant Creutzfeldt-Jakob disease (vCJD) isolates were used as seeds/inocula in both *in vitro* and *in vivo* propagation assays using the non-glycosylated human PrP$^C$-expressing mouse model (TgNN6h). After protein misfolding cyclic amplification (PMCA), all isolates maintained the biochemical characteristics of BSE. On bioassay, all PMCA-propagated BSE prions were readily transmitted to TgNN6h mice, in agreement with our previous *in vitro* results. TgNN6h mice reproduced the characteristic neuropathological and biochemical hallmarks of BSE, suggesting that the absence of glycans did not alter the pathobiological features of BSE prions. Moreover, back-passage of TgNN6h-adapted BSE prions to BoTg110 mice recovered the full BSE phenotype, confirming that the glycosylation of human PrP$^C$ is not essential for the preservation of the human transmission barrier for BSE prions or for the maintenance of BSE strain properties.

**Funding:** This work was supported financially by the following Spanish and European Interreg grants: RB, AGL2015-65560-R, MINECO Ministerio de Economía y Competitividad (Spanish Government) https://sede.mineco.gob.es/; RB, RTI2018-098711-B-I00, Ministerio de Ciencia, Innovación y Universidades (Spanish Government) https://www.ciencia.gob.es/; JC, RTI2018-098515-B-I00, Ministerio de Ciencia, Innovación y Universidades (Spanish Government) https://www.ciencia.gob.es/; JC, PID2021-122201OB-C21, Ministerio de Ciencia, Innovación y Universidades (Spanish Government) https://www.ciencia.gob.es/ partially supported by European Region Development Fund (ERDF) https://ec.europa.eu/regional_policy/en/funding/erdf/; RB & JJB, POCTEFA EFA148/16, European Region Development Fund (ERDF) https://ec.europa.eu/regional_policy/en/funding/erdf/. The funders had no role in study design, data collection and analysis, decision to publish, or preparation of the manuscript.

**Competing interests:** The authors have declared that no competing interests exist.

## Author summary

Bovine spongiform encephalopathy (BSE), publicly known as "mad cow disease", is a neurodegenerative disorder affecting cattle, caused by unconventional agents called prions. BSE can naturally transmit to human beings, producing the variant form of Creutzfeldt-Jakob disease (vCJD), which caused an unprecedented health and economic crisis in the UE. Prions are composed of PrP$^{Sc}$, a misfolded form of the cellular protein PrP$^C$, which can be variably glycosylated by conjugation with sugar molecules at two positions of its sequence. Several studies reported the role of PrP$^C$-attached sugars on important aspects of prion biology, such as the existence of different prion strains. Here, we demonstrate that it is possible to propagate BSE prions (from different animal and human sources) in a non-glycosylated human PrP$^C$ environment without loss of their strain properties. Different BSE isolates were successfully transmitted to a transgenic mouse model expressing non-glycosylated human PrP$^C$, and these animals manifested neuropathological and biochemical signs compatible with BSE. To definitely prove the maintenance of the strain, non-glycosylated BSE prions were transmitted to their original host: transgenic mice expressing cattle PrP$^C$. These animals recovered the full BSE phenotype, confirming that the glycosylation of human PrP$^C$ is not relevant for the propagation of this particular prion strain.

## Introduction

Prion diseases are fatal neurodegenerative disorders that include scrapie in sheep and goats, bovine spongiform encephalopathy (BSE) in cattle, chronic wasting disease (CWD) in cervids and Creutzfeldt-Jakob disease (CJD) in humans. Following its association with the variant form of Creutzfeldt-Jakob disease (vCJD) in humans [1], BSE represented one of the major public health crises in Europe in the last decades. Several studies strongly suggest that the BSE epidemic was caused by a single strain, which can be transmitted to a wide range of hosts without apparent alteration of its pathobiological features [2–6]. However, its pathogenicity towards humans can be enhanced through passages in other species, such as sheep and goats [7,8] or macaques [9].

A common feature of prion diseases is the accumulation of the pathological prion protein (PrP$^{Sc}$) in the central nervous system (CNS) of affected individuals. PrP$^{Sc}$ is a self-propagating, misfolded isoform of the host-encoded cellular prion protein (PrP$^C$), a membrane-anchored glycoprotein that is abundantly expressed in the CNS [10–12]. PrP$^C$ sequence contains two consensus sites for N-glycosylation, involving asparagine residues at positions 181 and 197 in the human PrP sequence (or corresponding positions in other species) which can be variably occupied [13], generating di-, mono-, and unglycosylated mature forms of PrP [14]. Several studies have demonstrated the key role of N-linked glycans in the intracellular trafficking and membrane location of PrP$^C$ [15–19]. This protein has a still-elusive function [20,21]. Although it was thought that the expression of PrP$^C$ in the cellular membrane was fundamental for the development of prion disease [22–25], it has been shown that mice expressing an anchorless PrP$^C$ (lacking the GPI attachment to the cell membrane) can develop a fatal transmissible amyloid encephalopathy [26]. Prion protein glycosylation can significantly modulate the interactions between heterologous PrP$^C$ and PrP$^{Sc}$ molecules, suggesting that glycans could be determining not only in the conversion efficiency of PrP$^C$ into its pathological counterpart, but also in the cross-species transmission of prions [27]. Other works using transgenic mice expressing an anchorless PrP$^C$, which due to altered post-translational processing is poorly

glycosylated, obtained similar results [28]. In addition, in the context of a defined host, PrP$^C$ can be misfolded into a great variety of prion strains, which are characterized by unique clinical features, neuropathological patterns and biological properties, including distinctive ratios of PrP$^{Sc}$ glycoforms. In general, strain properties are faithfully recapitulated upon serial passages in the same animal species [29–33]. Thus, the role of PrP$^C$ glycosylation in the transmission barrier phenomenon and in the encoding of strain-specific properties has been widely investigated *in vivo*.

In the present study, we assessed the behavior of BSE isolates with different transmission histories when propagated in a mouse model expressing non-glycosylated human PrP$^C$ (TgNN6h mice) [34], following both *in vitro* and *in vivo* approaches. Using protein misfolding cycling amplification (PMCA), all BSE prions propagated in the non-glycosylated human substrate. However, they showed different propagation efficiencies, in a way that was consistent with the existence of a transmission barrier. PMCA-propagated prions were readily transmitted to TgNN6h mice on bioassay, which developed biochemical and neuropathological hallmarks strongly indicative of BSE. Moreover, when TgNN6h-adapted BSE prions were back-passaged to a host expressing bovine PrP (BoTg110), the full BSE phenotype was recovered. Taken together, our results suggest that the absence of glycans alter neither the strength of the human transmission barrier for BSE nor the BSE strain pathobiological features.

## Results

### Non-glycosylated human PrP substrate is converted *in vitro* by classical BSE and BSE-derived isolates

In order to assess the *in vitro* misfolding ability of non-glycosylated human PrP and how the lack of glycans could affect the human transmission barrier for BSE prions, TgNN6h mice brain homogenates were seeded *in vitro* with BSE isolates of different origins: classical cattle BSE (BSE), sheep (sBSE) and pig-passaged BSE (pBSE) [35] and human vCJD. As mentioned before, classical BSE prions present particular abilities to spread to distinct animal species [36–38], including humans [1], and, moreover, they show singular stable pathobiological features [2,4,5].

Each BSE isolate was subjected, in quadruplicate, to 15 serial PMCA rounds in TgNN6h substrate to compare their abilities to induce the misfolding of non-glycosylated human PrP$^C$. The vCJD isolate readily propagated in TgNN6h substrate with a 100% efficacy (4/4 positive replicates) after a single 24 h round. In contrast, cattle BSE, sheep-passaged BSE, and pig-passaged BSE required more than one PMCA rounds to propagate, and they did so with different efficiencies (Fig 1A), in a way consistent with the existence of a transmission barrier. In particular, BSE required 15 rounds for a 25% amplification, and sBSE needed 7 rounds to show 75% propagation, whereas pBSE reached 100% propagation efficiency within 9 rounds. The fact that BSE propagated significantly worse than sBSE (and pBSE) in human substrate likely correlates with its more difficult transmission capability *in vivo* when compared to sheep-passaged BSE, reported elsewhere [8].

As expected, given the absence of different glycoforms in TgNN6h PrP$^C$, PMCA products derived from the propagation of all four BSE sources on TgNN6h substrate showed a PrP$^{Sc}$ signature characterized by a single, non-glycosylated band at 19 kDa (Fig 2). Given that PMCA is a highly stochastic technique (the efficiency among rounds can differ significantly), and that these samples were submitted for SDS-PAGE and Western blot without prior adjustment of quantities, the differences in signal intensity observed among different TgNN6h-propagated isolates are likely due to dissimilarities in PrP$^{Sc}$ amount rather than to different resistances to PK.

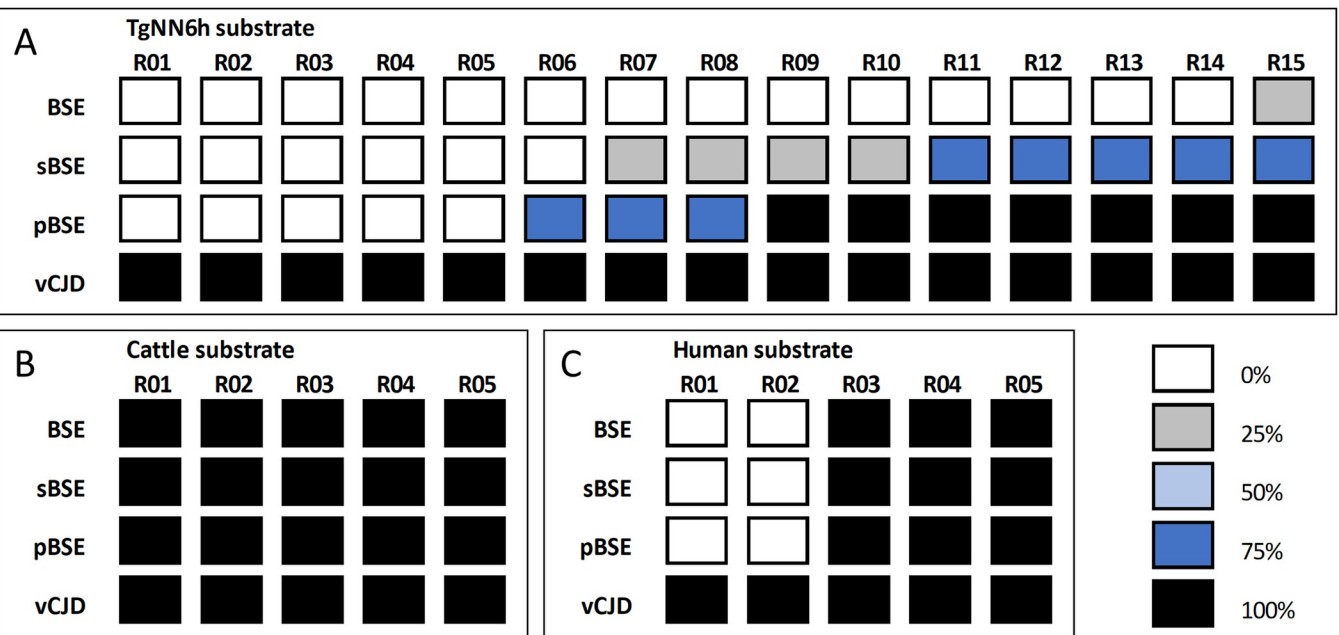

**Fig 1.** *In vitro* **propagation assay of different BSE sources on TgNN6h substrate (A), normal cattle brain substrate (B) and normal human brain substrate (C).** The color scale represents the proportion of positive tubes (showing proteinase K-resistant PrP by Western blot) out of the total number of tubes subjected to PMCA (n = 4).

As predicted from previous studies demonstrating the absence of transmission barrier of BSE prions towards its natural host, all isolates propagated in just one 24-h PMCA round in wild-type cattle brain substrate (Fig 1B). In contrast, the distinct BSE isolates showed different propagation efficiencies when using normal human brain homogenate as substrate. The vCJD inoculum propagated in all replicates in the first round, whereas BSE, sBSE and pBSE isolates did not overcame the barrier until round 3 (Fig 1C). Again, this result agreed with the presence of a transmission barrier towards humans and suggests that the difficulties encountered with the TgNN6h substrate for propagating BSE prions is likely unrelated to its glycosylation status, but rather linked to the human PrP$^C$ sequence.

The products obtained from PMCA propagation on either cattle or human substrate showed the PrP$^{Sc}$ signature characteristic of BSE, i.e. the non-glycosylated band at 19 kDa and a predominance of the diglycosylated species (Fig 2). This was true also for pBSE, whose original isolate presented, at variance with the rest of BSE sources, a banding pattern characterized by a predominance of the monoglycosylated band; this is likely a particularity of porcine PrP [35,39] and does not entail a modification of the BSE strain contained in this inoculum. In contrast, cattle BSE, sBSE and vCJD original isolates presented the prototypic BSE banding pattern from the beginning.

We further propagated the four BSE inocula in TgNN6h substrate for 10 additional PMCA rounds (to ensure the absence of infectivity coming from the original sources) and subjected them to *in vivo* bioassay. The PMCA-adapted isolates were termed BSE-PMCA, sBSE-PMCA, pBSE-PMCA and vCJD-PMCA.

## PMCA propagation of BSE isolates greatly facilitated the transmission to TgNN6h mice while direct inoculation required longer incubation times

The original inocula (consisting of 1% brain homogenates) and the *in vitro*-adapted inocula (following sufficient serial PMCA passages to eliminate any traces of the original seed) were

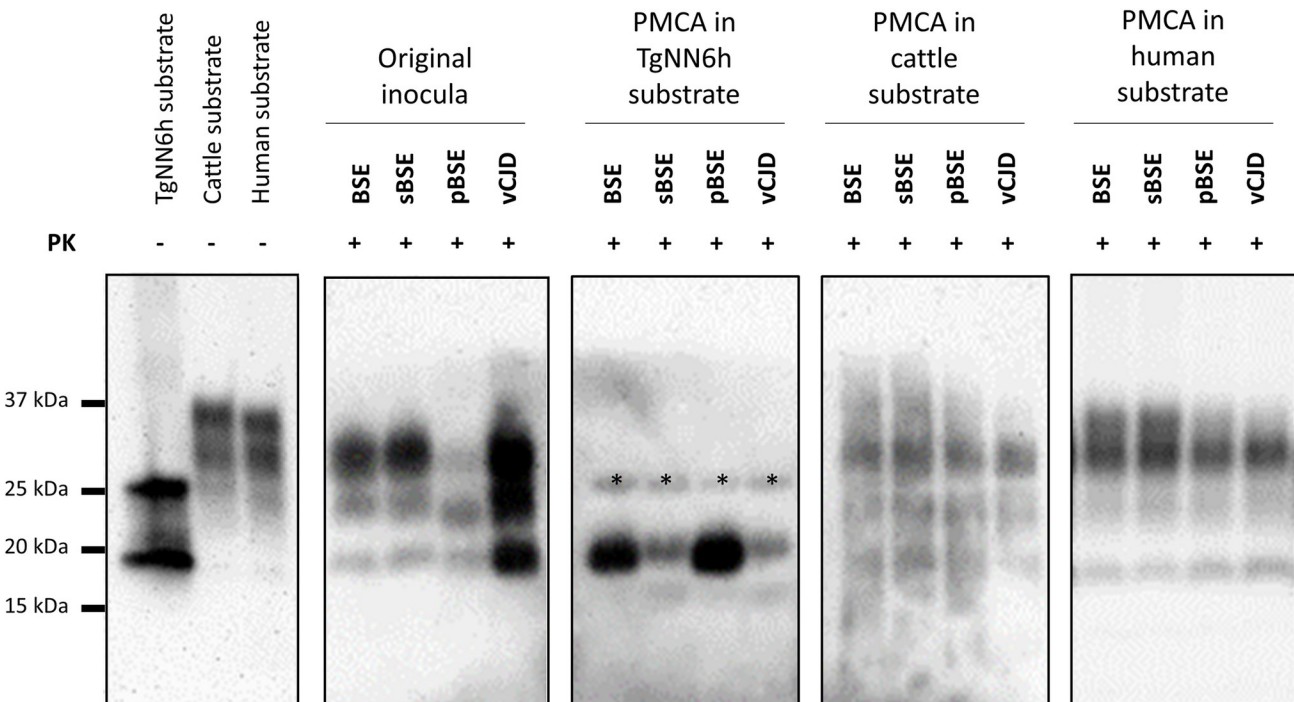

**Fig 2. Biochemical analysis of PMCA-propagated BSE prions on TgNN6h and normal cattle and human brain substrates.** Original inocula: brain homogenates from BSE-infected cattle, BSE-infected sheep (sBSE), BSE-infected pig (pBSE), or a vCJD patient; PMCA in TgNN6h substrate: isolates generated *in vitro* after 19 rounds of PMCA in TgNN6h substrate; PMCA in cattle substrate: isolates generated *in vitro* after 13 rounds of PMCA in wild-type bovine substrate; PMCA in human substrate: isolates generated *in vitro* after 16 rounds of PMCA in wild-type human substrate. Original inocula and samples propagated in cattle and human substrate were digested with 85 µg/mL of proteinase K (PK), while TgNN6h-propagated samples were digested with 170 µg/mL PK, and analyzed by Western blot using monoclonal antibody 6H4 (1:10,000); bands corresponding to incomplete digestion are marked with asterisks. Undigested TgNN6h, cattle, and human substrates were loaded as controls.

inoculated in TgNN6h mice. This transgenic line expresses non-glycosylated human 129M PrP$^C$ at approximately 60% of the physiological levels [34]. The original, non-adapted isolates were also bioassayed in Tg340 mice, which overexpresses a fully glycosylated human 129M PrP$^C$ (at levels 4-fold higher than those detected in human brain), as controls [7]. At least 6 animals of either transgenic line were inoculated with each of the isolates (Table 1).

TgNN6h mice inoculated with the *in vitro*-propagated BSE-PMCA, sBSE-PMCA, pBSE-PMCA and vCJD-PMCA isolates succumbed to prion disease on first passage with a 100% attack rate. TgNN6h animals that developed clinical disease showed hyperesthesia, kyphosis and ataxic gait in a first stage of the disease, followed by weight loss, lethargy and ruffled coat in a later phase. These transmissions to TgNN6h mice occurred with similar survival periods. Second passage of BSE-PMCA, sBSE-PMCA and pBSE-PMCA isolates resulted also in 100% attack rates in TgNN6h mice in all cases, with survival periods similar to those seen in the first passage (Table 1).

Notably, no TgNN6h mouse inoculated with the direct (i.e. not PMCA-propagated) BSE, sBSE, or pBSE isolates developed disease or accumulated detectable levels of PrP$^{Sc}$ in the brain for up to 600 dpi (Table 1); this was expected given the low PrP$^C$ expression level of these animals (0.6x) and agrees with studies in which BSE was unable to infect knock-in models (thus expressing physiological levels of the protein) [40]. However, after 584 dpi, one animal from the TgNN6h group challenged with the direct vCJD inoculum developed neurological signs, particularly tremor and ataxic gait. This animal and five other vCJD-inoculated TgNN6h mice

**Table 1. Inoculation of TgNN6h and Tg340 mice with BSE (and BSE-derived) prions.**

| Isolates | TgNN6h mice (1st passage) | | TgNN6h mice (2nd passage) | | Tg340 mice (1st passage) | |
|---|---|---|---|---|---|---|
| | Survival time (dpi) (mean ± SEM)[a] | Attack rate[b] | Survival time (dpi) (mean ± SEM)[a] | Attack rate[b] | Survival time (dpi) (mean ± SEM)[a] | Attack rate[b] |
| BSE | 127–733 | 0/11 | 161–476 | 0/4 | 488–711 | 0/6 |
| sBSE | 138–641 | 0/11 | 132–539 | 0/5 | 700–756 | 0/6 |
| pBSE | 139–648 | 0/9 | 179–546 | 0/6 | 735[c] | 1/7 |
| vCJD | 697±23 | 6/11 | 588±52 | 2/5 | 616±57 | 4/5 |
| BSE-PMCA | 279±24 | 11/11 | 239±5 | 6/6 | - | - |
| sBSE-PMCA | 222±24 | 7/7 | 228±6 | 7/7 | - | - |
| pBSE-PMCA | 206±32 | 11/11 | 238±4 | 5/5 | - | - |
| vCJD-PMCA | 206±46 | 6/6 | - | - | - | - |

[a] Survival times are shown as mean number of days between inoculation and euthanasia ± SEM, except when none of the inoculated mice developed clinical signs consistent with a TSE, or were found to be PrP$^{Sc}$ positive, in which case the survival periods of the first and last dead animal in each group are presented

[b] Data based on PrP$^{Sc}$ detection in the brain.

[c] All animals were sacrificed without clinical signs at this age, and only one turned out positive by Western blot.

dpi: days post-inoculation, SEM: standard error of the mean.

that were culled at ∼700 dpi with no signs of clinical disease presented PrP$^{Sc}$ accumulation in the brain (Table 1).

At first passage, none of the Tg340 inoculated mice developed disease after the inoculation of BSE, sBSE, or pBSE isolates. However, one pBSE-inoculated Tg340 mice culled at 735 dpi was positive for PrP$^{Sc}$ in the brain by Western blot. In contrast, vCJD transmitted to Tg340 mice, leading to incomplete attack rates and long survival periods (Table 1), indicative of a considerable transmission barrier. These results, although contrasting with those published elsewhere [7], were expected since the inoculations were performed with 1% brain homogenates, i.e. 10 times less infectious material than in most other studies, that use 10% brain homogenates. In addition, the BSE and vCJD isolates used were different from those inoculated in the present study, and therefore it is possible that these isolates contained different levels of infectivity. This, together with the prolonged incubation times close to the lifespan of the animals observed also in previous studies, could explain the absence of transmission in our case.

All TgNN6h animals that accumulated PrP$^{Sc}$ in the brain presented a banding pattern characterized by a clear band with a molecular weight of 19 kDa, likely corresponding to non-glycosylated BSE PrP$^{Sc}$. A lower band (around 15 kDa) was consistently observed only in these animals (Fig 3), but not in Tg340 mice. Deglycosylation of BSE-infected TgNN6h samples with PNGase F did not provoke a shift of this band (S1 Fig). In addition, Tg340 samples treated with PNGase F showed a reduction of the three-bands pattern to a single 19–20 kDa band, but it did not show the 15 kDa band (S1 Fig). Therefore, we believe that this band corresponds to an endoproteolytic fragment specific to TgNN6h mice, although the reason why it only arises from unglycosylated PrP$^{Sc}$ remains to be investigated.

In contrast, Tg340 mice accumulating PrP$^{Sc}$ in brain after infection with pBSE and vCJD replicated the prototypical BSE banding pattern with predominance of the diglycosylated species, as expected (Fig 3).

## Common neuropathological features in TgNN6h mice infected with PMCA-adapted BSE prions suggest that they are the same strain

TgNN6h mice challenged with BSE-PMCA, sBSE-PMCA, pBSE-PMCA and vCJD-PMCA developed very similar neuropathological features. The brain of these mice showed very

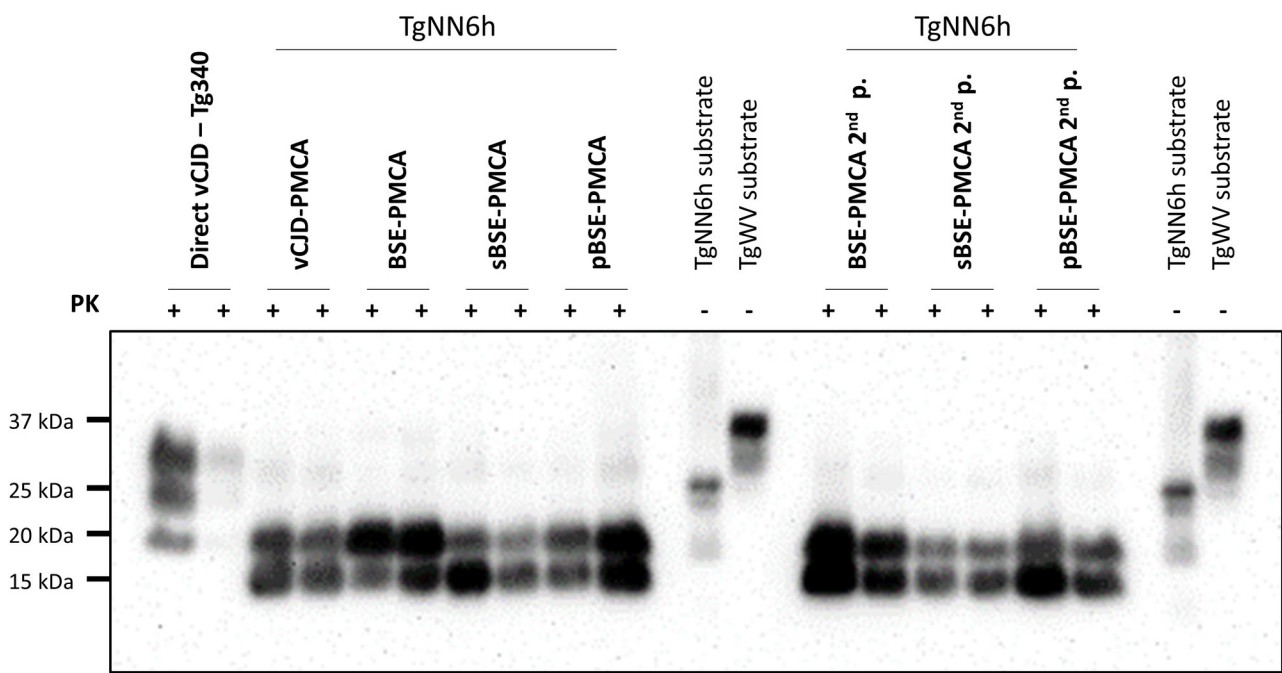

**Fig 3. PrP^Sc detection from first and second-passage BSE-PMCA, sBSE-PMCA, pBSE-PMCA, and vCJD-PMCA inoculated TgNN6h mice, as well as direct vCJD-inoculated Tg340 mice.** 10% brain homogenates from challenged mice were digested with 170 μg/mL of proteinase K and analyzed by Western blot using 3F4 antibody (1:10000). Undigested 10% brain homogenate from TgNN6h (non-glycosylated human PrP^C) and TgWV (normally glycosylated human PrP^C) were included as controls.

intense spongiosis and abundant amyloid-like plaques surrounded by areas of severe spongiform change (florid plaques). Spongiform lesions were particularly intense in brain cortex and diencephalon, especially in the thalamus. Other brain areas, such as brainstem and cerebellum, presented scarcer spongiform changes (Fig 4).

Immunohistochemical analyses revealed that the plaques that were observed with hematoxylin and eosin staining were composed of prion protein, presenting a PrP amyloid core (confirmed by Congo red staining, as showed in S2 Fig) generally surrounded by a halo of spongiform degeneration (Fig 4B). These neuropathological changes are termed florid plaques and are characteristic of vCJD in humans [41,42]. Large florid plaques were detected in the cerebral cortex, thalamus, hypothalamus, and white matter structures such as the corpus callosum. In certain brain areas, especially in the hippocampus, plaques were usually confluent and very disruptive. In other areas, such as cerebellum and medulla oblongata, amyloid plaques were more discretely distributed. In addition to plaques, granular deposits of prion protein were detected throughout the brain. No differences in neuropathological characteristics were observed among the different *in vitro*-adapted BSE sources or between the first and second passage of TgNN6h mice.

The histopathological study of those Tg340 mice successfully infected with pBSE or vCJD revealed the presence of florid plaques in the subcallosal area of these mice (S3 Fig). The coincidence on these neuropathological features with those found in TgNN6h suggests that the absence of PrP glycosylation in TgNN6h did not alter the characteristic neurotropism of the BSE strain.

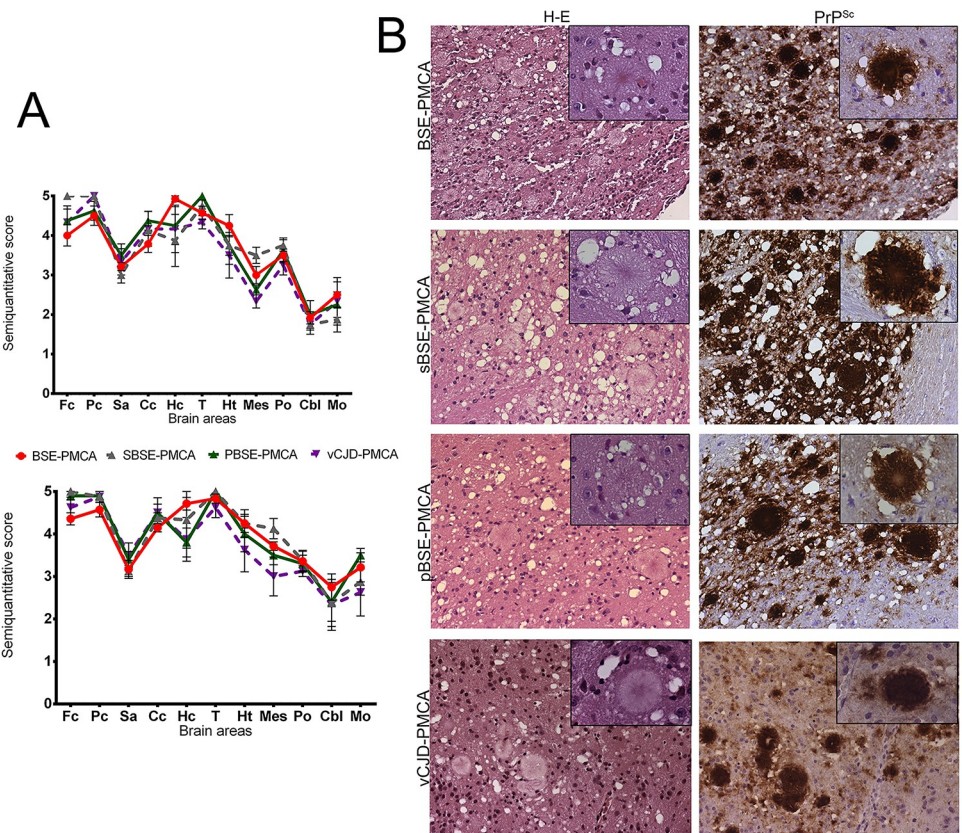

**Fig 4. Neuropathological features of TgNN6h mice inoculated with PMCA-propagated BSE prions. A** Spongiosis and PrP<sup>Sc</sup> deposition profiles. Spongiform lesions and PrP<sup>Sc</sup> deposition were evaluated semiquantitatively on a scale of 0 (absence of lesions/deposits) to 5 (high intensity lesion/deposition) in the following brain areas: frontal cortex (Fc), parietal cortex (Pc), septal area (Sa), corpus callosum (Cc), hippocampus (Hc), thalamus (T), hypothalamus (Ht), mesencephalon (Mes), pons (Po), cerebellum (Cbl), and medulla oblongata (Mo). All BSE inocula showed almost identical neuropathological profiles when transmitted to TgNN6h mice. **B** Hematoxylin and eosin staining and immunohistochemical analysis of the brains of BSE-PMCA, sBSE-PMCA, pBSE-PMCA or vCJD-PMCA affected mice showed the presence of conspicuous plaque-like deposits, surrounded by areas of severe spongiform change (florid plaques). Scale: x20, insert pictures: x60. Immunohistochemistry was performed using the 3F4 antibody (1:1,000). Brain areas: Parietal cortex (BSE-PMCA), mesencephalon (sBSE-PMCA), and thalamus (pBSE-PMCA and vCJD-PMCA).

## The recovery of strain properties after transmission of TgNN6h-adapted BSE prions to BoTg110 mice suggests the maintenance of the BSE strain

To check whether BSE characteristics were maintained after serial transmission in a human non-glycosylated PrP environment, TgNN6h-adapted isolates were transmitted to transgenic mice overexpressing cattle PrP. Brains from second-passage TgNN6h mice inoculated with the PMCA-adapted BSE, sBSE and pBSE isolates (hereafter referred to as BSE-TgNN6h, sBSE-TgNN6h and pBSE-TgNN6h) were inoculated into BoTg110 mice, which express bovine PrP<sup>C</sup> at levels 8-fold those of cattle brain. As a control, a group of BoTg110 mice was infected with a natural cattle BSE isolate.

While cattle BSE readily transmitted to BoTg110 mice (100% attack rate) and incubation periods of around 300 dpi, as previously reported [4,7], all three non-glycosylated isolates propagated with incomplete attack rates and longer and heterogeneous incubation periods (Table 2). These discrepant results are unlikely to reflect a change in the strain properties of BSE prions upon passage in the TgNN6h model (since, as stated below, the biochemical and

neuropathological hallmarks were identical to those obtained with the original cattle BSE isolate). Rather, they evidence the transmission barrier naturally occurring between cattle and human, likely increased by the two additional amino acid changes introduced in the TgNN6h PrP$^C$ sequence to impede glycosylation.

Two additional factors may have contributed to this reduced transmission efficiency: 1) the fact that TgNN6h mouse brains express ~40% less PrP$^C$ than a human brain, thus accumulating lower amounts of PrP$^{Sc}$ and infectivity, and 2) the use of 1% (instead of the standard 10%) brain homogenates as inocula, which further reduces the infective dose administered to each animal. In order to prove this hypothesis, we performed an *in vitro* titration experiment. Both the original cattle BSE isolate and the BSE-TgNN6h, sBSE-TgNN6h, and pBSE-TgNN6h inocula that were used to infect BoTg110 mice were serially diluted and submitted to two serial PMCA rounds in BoTg110 substrate. While cattle BSE was able to propagate up to dilution $10^{-5}$, all of the TgNN6h-passaged isolates propagated only up to dilution $10^{-1}$ (S4 Fig). These results allowed us to conclude that the propagative dose that accumulates in the brain of TgNN6h mice is significantly lower than that of BSE-infected cattle, thus explaining the low transmission efficiency of these isolates in first-passage BoTg110 mice.

BoTg110 mice inoculated with these non-glycosylated BSE isolates developed biochemical and neuropathological features identical to those observed in BoTg110 mice inoculated with cattle BSE. In contrast to TgNN6h mice, which showed a single PrP$^{Sc}$ band on Western blot, BoTg110 mice challenged with BSE-TgNN6h, sBSE-TgNN6h and pBSE-TgNN6h inocula displayed the full, three-banded PrP$^{Sc}$ profile characteristic of BSE, i.e. non-glycosyated band at 19 kDa and a predominance of the diglycosylated band (Fig 5).

In addition, these mice showed widespread spongiform degeneration, especially prominent in septal area, thalamus, medulla oblongata and pons, with lower scores in hippocampus, mesencephalon, and cerebellar cortex. This distribution of neuropathological lesions is very similar to that observed by other authors in BoTg110 mice inoculated with BSE [43,44]. All BoTg110 mice that succumbed to disease developed fine-punctate, granular and plaque-like deposits, following a similar brain distribution (Fig 6). This morphological pattern has been described before in the same model inoculated with BSE isolates [4,43].

## Discussion

In the present study, we aimed at assessing whether the absence of glycans in the human PrP$^C$ could impact the transmission barrier and the strain properties of BSE prions. BSE is the only animal prion strain for which a natural transmission to humans has been described, leading to

**Table 2. Inoculation of BoTg110 mice with non-glycosylated BSE isolates and cattle BSE.**

| Isolates | BoTg110 mice | |
|---|---|---|
| | Survival time (dpi) (mean ± SEM)[b] | Attack rate[c] |
| BSE-TgNN6h[a] | 412±11 | 3/6 |
| sBSE-TgNN6h[a] | 430±7 | 4/5 |
| pBSE-TgNN6h[a] | 393±4 | 3/6 |
| Cattle BSE | 312±6 | 7/7 |

[a] BSE-TgNN6h, sBSE-TgNN6h, pBSE-TgNN6h: brain homogenates from the second passage in TgNN6h mice of cattle, sheep-passaged and porcine-passaged BSE adapted *in vitro* to non-glycosylated human substrate.

[b] Survival times are shown as mean number of days between inoculation and euthanasia ± SEM.

[c] Data based on PrP$^{Sc}$ detection in the brain.

dpi: days post-inoculation, SEM: standard error of the mean.

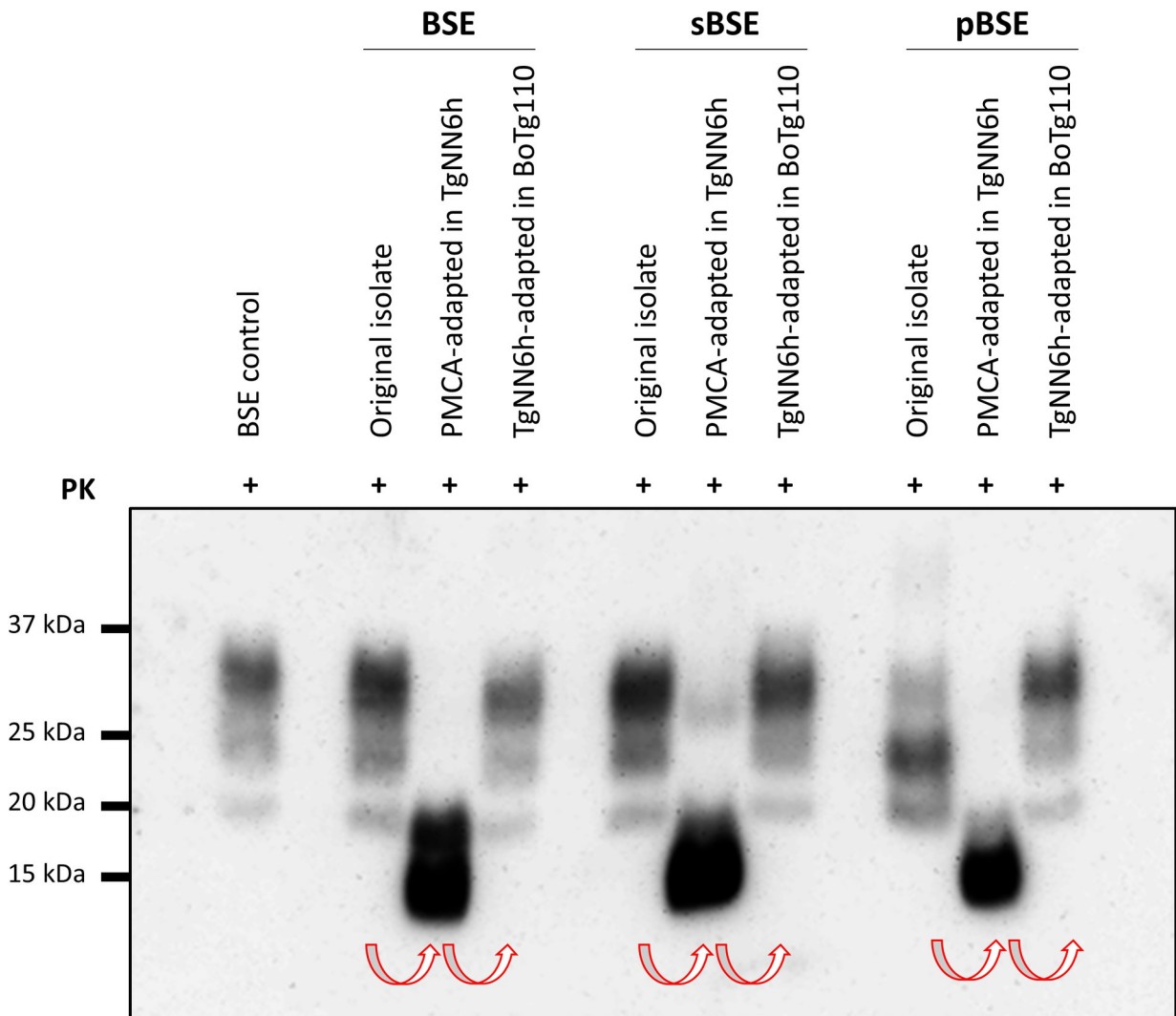

**Fig 5. Recovery of BSE banding pattern on passage from TgNN6h to BoTg110 of BSE, sBSE and pBSE isolates.** Brain homogenates at 1% from challenged mice were digested with 85 µg/mL (for original isolates) or 170 µg/mL (for TgNN6h and BoTg110 brains) of proteinase K and analyzed by Western blot using 6H4 antibody (1:10,000). A transition from a classical three-banded BSE pattern in the original isolates to a single 19-kDa band-containing pattern in TgNN6h mice was observed, followed by a complete recovery of the three-banded pattern in BoTg110 mice. Note that the banding pattern of the pBSE original isolate differs from the classical BSE pattern in being predominantly monoglycosylated (a hallmark imposed by porcine PrP$^C$), while after passage to BoTg110 the pattern is identical to those of other BSE sources. Red arrows indicate passage history.

its human counterpart variant Creutzfeldt-Jakob disease (vCJD). Prions accumulating in the brain of vCJD patients maintain BSE pathobiological features [1,45,46]. Aside from classical BSE, the potential cross-species transmission of prions to humans has been demonstrated for classical scrapie [47], CWD [48], and L-BSE [49–51] by experimental challenge of transgenic mice overexpressing human PrP$^C$ and/or *in vitro* propagation techniques. However, these successful transmissions are the exception rather than the rule since many other similar studies have reported opposite results [48,52–54]. Indeed, in parallel experiments (not shown in this publication) we seeded the TgNN6h substrate with SSBP/1, atypical scrapie, CWD, L-BSE and H-BSE isolates and subjected them to 15 rounds of serial PMCA. None of these prion strains was able to propagate in this non-glycosylated human substrate. TgNN6h mice intracerebrally

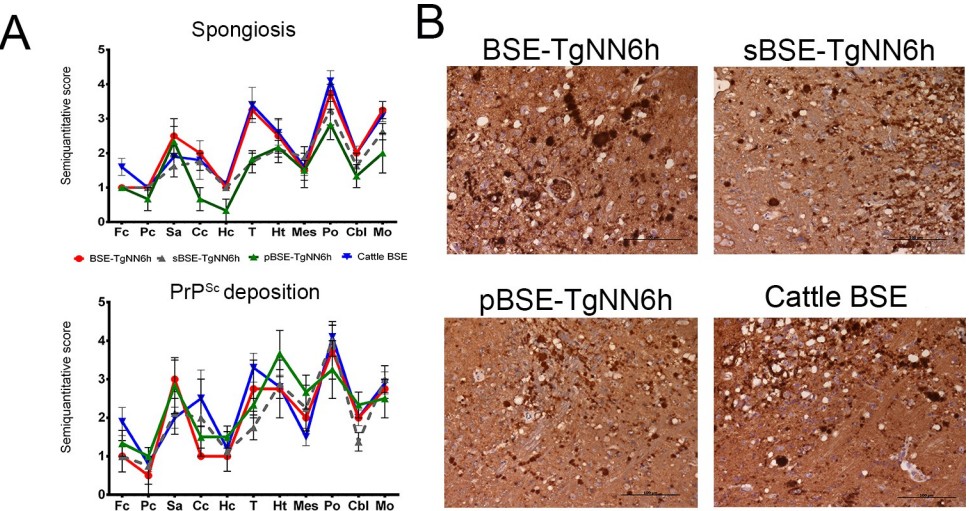

**Fig 6. Neuropathological features of BoTg110 mice inoculated with non-glycosylated BSE isolates and normally glycosylated cattle BSE. A** Spongiosis and PrP$^{Sc}$ deposition profiles. Spongiform lesions and PrP$^{Sc}$ deposition were evaluated semiquantitatively on a scale of 0 (absence of lesions/deposits) to 5 (high intensity lesion/deposition) in the following brain areas: frontal cortex (Fc), parietal cortex (Pc), septal area (Sa), corpus callosum (Cc), hippocampus (Hc), thalamus (T), hypothalamus (Ht), mesencephalon (Mes), pons (Po), cerebellum (Cbl), and medulla oblongata (Mo). Non-glycosylated isolates generated very similar neuropathological features to those of cattle BSE when transmitted to BoTg110 mice. **B** Immunohistochemical analysis of the brains of BoTg110 mice inoculated with the different isolates (pons). All inocula produced intense vacuolation and PrP$^{Sc}$ plaques especially intense at the pons level. Immunohistochemistry was performed using the 6H4 antibody (1:1,000).

challenged with the same strains did not succumb to disease or accumulate PrP$^{Sc}$ in their brains.

In our study, all BSE isolates propagated in the TgNN6h substrate. The vCJD isolate showed the highest efficiency on *in vitro* propagation (Fig 1A) and, in correlation with this, it was the only one able to transmit *in vivo* after the inoculation of the original isolate (Table 1). This behavior parallels the results obtained when the same BSE isolates were propagated in wild-type human substrate (Fig 1B), suggesting that the absence of glycans in TgNN6h PrP$^C$ does not alter the human transmission barrier for BSE prions. The higher efficiency of vCJD with respect to other BSE prions for propagating in wild-type human substrate has been previously reported [48]. Since PMCA was designed as a methodology to accelerate the misfolding process [55], it is conceivable that the compatibility of amino acid sequences between seed and substrate and other slight adaptations of vCJD to the human brain environment accounts for the high propagation efficiency in human substrate observed for this particular isolate [45]. In addition, transmission studies in humanized transgenic mice and primates have demonstrated that BSE, although showing very stable pathobiological features upon transmission, can readily adapt to the new host [9,40], and once transmitted to human beings in the form of vCJD, the barrier for human-to-human transmission is substantially reduced [40]. This fact could explain the results obtained in the TgNN6h mouse bioassay, since vCJD was the only isolate that transmitted to mice at first passage after the direct inoculation of the original source (Table 1).

The propagation efficiency of all BSE sources was absolute in the bovine substrate (Fig 1C). BSE-derived prions, irrespective of their original host (cattle, sheep, pig or human) transmit to transgenic mice expressing bovine PrP$^C$ with no differences in regard to survival times or pathobiological characteristics, indicating a lack of a transmission barrier for the propagation of BSE prions in a bovine PrP substrate [38]. However, they transmit poorly to transgenic

human models [38,40,50,56], pointing to the existence of a strong transmission barrier between cattle and humans, even though the transmission of BSE to humans naturally occurs producing vCJD [1].

Among BSE isolates sourced from animal species, we observed that both sBSE and pBSE isolates propagated in TgNN6h substrate more efficiently than cattle BSE (Fig 1A). These results are in accordance with previous studies showing that experimental sBSE prions propagate more efficiently than cattle BSE in transgenic mice expressing human $PrP^C$ [7,57], which was attributed to a better structural compatibility between sheep $PrP^{Sc}$ and human $PrP^C$ [7]. However, we did not observe the same behavior with the wild-type human substrate. The reason for this could be related to the higher $PrP^C$ expression levels of human brain in comparison with TgNN6h, which, in combination with the extraordinary sensitivity of PMCA [55,58] make these differences undetectable *in vitro*.

The implication of the glycosylation status of $PrP^C$ on intra- and inter-species transmission of prion strains has been discussed at length, and several studies have suggested that glycosylation of $PrP^C$ could be a key factor influencing transmission barrier [15,27,59,60]. Wiseman and collaborators showed that transgenic mice expressing non-glycosylated murine $PrP^C$ (G3 mice) were completely resistant to prion disease, and that this alteration was associated to the absence of the first glycosylation site since the elimination of the second site resulted in the efficient transmission of human prions. [60]. As aforementioned, we did not observe any significant alterations of the cross-species transmission barrier for BSE prions in the TgNN6h non-glycosylated human model. Differences in the isolates and/or the transgenic models used could explain the discrepancies between our results and those obtained with the G3 transgenic line [60]. Nevertheless, it is also possible that glycans affect prion propagation in a strain or species-dependent manner. BSE is a particular strain, with the ability of propagating in $PrP^C$ from different species without significantly altering its features [6].

All TgNN6h mice inoculated with the *in vitro*-propagated BSE-PMCA, sBSE-PMCA, or pBSE-PMCA isolates succumbed to prion disease in ∼200 dpi and proved positive for $PrP^{Sc}$ accumulation by immunohistochemical techniques and Western blot (Figs 3 and 4). Animals inoculated with direct brain homogenates BSE, sBSE and pBSE did not develop disease in more than 700 dpi. These results corroborate that the direct inoculation requires incubation times longer than the normal lifespan of these mice (especially considering that inoculation were performed with 1% brain homogenates and that this model expresses human $PrP^C$ at levels of only 0.6x), while PMCA serial propagation in TgNN6h substrate greatly facilitates overcoming the transmission barrier by the progressive stabilization and adaptation of the *in vitro* generated prions [61].

Mice inoculated with BSE-PMCA, sBSE-PMCA, pBSE-PMCA and vCJD-PMCA presented very severe neuropathological changes, and showed abundant deposits with the characteristics of florid plaques described in vCJD-affected patients and humanized models (Fig 4) [7,42]. Unglycosylated prions have been reported to favor extracellular plaque formation in murine models of prion disease, in which these prions colocalize with plaque deposits [62]. These results, together with the results obtained in the present study, suggest that florid plaques observed in humanized mice and humans are preferentially formed from unglycosylated $PrP^{Sc}$. Our PMCA-propagated inocula were very similar with respect to the distribution of neuropathological changes (Fig 3), which also coincided with those previously described for Tg340 [7] and Tg650 [63] mice (expressing fully glycosylated human $PrP^C$) inoculated with BSE and BSE-related prions. Overall, our results indicate that the neuropathological hallmarks of BSE were maintained after transmission to a humanized non-glycosylated host.

When analyzed by Western blot, TgNN6h mice that developed disease displayed a single unglycosylated band with a molecular weight of 19 kDa (Fig 3), in most cases accompanied by

a low molecular weight fragment, likely a consequence of endoprotease activity on the samples. This lower fragment can be also observed in undigested TgNN6h substrate (Fig 2), but not in Tg340 mice inoculated with vCJD, not even after deglycosylation (S1 Fig). Although we did not further characterized this 15 kDa band seemingly characteristic of TgNN6h mice, we believe that it may correspond to an endoproteolytic fragment similar to others previously described [64], and that apparently occur preferentially on unglycosylated substrate. This glycoprofile was almost identical to that displayed by *in vitro*-propagated seeds in TgNN6h substrate (Fig 2). These biochemical features are suggestive of BSE [65], indicating that glycans are likely not necessary to maintain the pathobiological features of classical BSE prions. This was later unequivocally proved by the observation that BoTg110 inoculated with TgNN6h-passaged BSE prions developed a prion disease whose neuropathological hallmarks and biochemical features fully coincided with those of BoTg110 animals infected with reference BSE isolates in this study and others [4,7]. These TgNN6h-passaged BSE prions, however, transmitted with incomplete attack rates to BoTg110 mice, in contrast with the results obtained by Padilla and colleagues with normally glycosylated BSE prions. These discrepant results could be explained by the transmission barrier existing between human and bovine PrP sequences, likely increased due to the presence of two additional asparagine-to-glutamine substitutions in the TgNN6h model to avoid glycosylation. Also, the fact that TgNN6h mouse brains contain ~40% less $PrP^C$ than human brain could lead to a lower accumulation of $PrP^{Sc}$ and infectivity in this model, and the use of 1% brain homogenates for inoculation, instead of the 10% used in previous studies, could contribute to a reduction in the infective dose administered to each BoTg110 animal. A second passage in the same animal model would definitely demonstrate that the incomplete attack rates observed in first-passage BoTg110 animals are due to the existence of such transmission barrier and/or to reduced infectivity accumulating in TgNN6h brains, rather than to an alteration in the strain features of BSE upon passage in the unglycosylated model. In fact, neuropathological features in BoTg110 mice inoculated with the TgNN6h-passaged BSE prions were very similar from those reported by us and other authors in BoTg110 mice inoculated with fully glycosylated BSE prions, regarding distribution [43,44] and morphology of $PrP^{Sc}$ deposits [4,43]. Therefore, a second passage in BoTg110 was deemed unnecessary to prove that propagation upon an unglycosylated $PrP^C$ does not alter BSE strain features. Nonetheless, in order to confirm that TgNN6h mouse brains accumulate less infectivity with respect to the classical BSE isolate from cattle, an *in vitro* titration experiment was performed by submitting serial dilutions of all the inocula to two serial PMCA rounds in BoTg110 substrate. As shown in S4 Fig, original cattle BSE isolate was detectable up to dilution $10^{-5}$ in the second PMCA round, whereas the three inocula derived from TgNN6h brain homogenates were able to propagate only up to dilution $10^{-1}$, i.e. a 10,000-fold lower propagation capability. This apparently lower titer, together with a transmission barrier and the long incubation times characteristic of this strain, which are close to the lifespan of the animals, can explain the incomplete attack rates observed *in vivo*.

Several studies have suggested that the glycosylation status of host $PrP^C$ could strongly determine the phenotypic characteristics of the infecting strain [66] and the transmission efficiency of prions between different species [60]. However, these effects have been observed with some strains but not with others, which indicates that glycans may not be essential for the retention of strain-specific properties [66], or that the strains present dramatically different requirements with respect to the glycosylation status of host $PrP^C$. Piro *et al.*, 2009 showed that unglycosylated $PrP^{Sc}$ molecules, generated *in vitro* using an enzymatically deglycosylated mouse $PrP^C$ as a substrate, maintained their strain-dependent neuropathological and biochemical features when inoculated in wild-type mice. These results led to the hypothesis that unglycosylated $PrP^{Sc}$ molecules can encode strain-specific patterns of $PrP^{Sc}$ accumulation

[67]. This conclusion was further supported by Moudjou *et al.*, 2016 by the propagation of 127S scrapie prions in ovine PrP glycosylation mutants and their subsequent transmission to Tg338 ovinized mice. Non-glycosylated, PMCA-propagated 127S prions reproduced the neuropathological and biochemical features of normally glycosylated 127S prions when inoculated in Tg338 mice. In addition, these unglycosylated prions recovered the three-band pattern when propagated in wild-type PrP$^C$ by PMCA. Thus, they concluded that glycans do not play a major role in determining strain-specific properties, and that these are encoded in the structural backbone of PrP$^{Sc}$ [68].

In line with this, we observed that TgNN6h mice that developed the disease after transmission of BSE isolates showed similar pathobiological features to those described in natural cases of human transmissions [42,65] and in BSE/vCJD-challenged transgenic mice expressing fully glycosylated human PrP$^C$ [7,63]. In addition to this, the back-passage of these unglycosylated prions in their original PrP$^C$ environment, i.e. BoTg110 mice expressing wild-type cattle PrP$^C$, resulted in the recovery of the BSE full phenotype. BoTg110 mice were chosen for this study as they transmit the BSE agent in absence of any transmission barrier, and are thus the most suitable model to assess whether unglycosylated BSE prions recover their prototypal characteristics. Alternatively, a back-passage of the TgNN6h-propagated BSE isolates into humanized, instead of bovinized, transgenic mice would also provide interesting clues on the properties of these unglycosylated BSE prions. Overall, thus, our results suggest that glycans are not vital on the determination of this transmission barrier or for the conservation of the pathobiological features of BSE prions.

## Materials and methods

### Ethics statement

All animal experiments were approved by the Ethics Committee for Animal Experiments of the University of Zaragoza (permit number PI20/15) and were carried out in accordance with the recommendations for the care and use of experimental animals and in agreement with Spanish law (R.D. 1201/05).

Biological material from human origin was acquired from the BioBank of Fundación Hospital Alcorcón de Madrid, Madrid, Spain, under agreement 3/2013. The transfer and use of this sample was approved by the external Bioethics Committee and internal Technical Committee from the BioBank, under Spanish Biomedical Research law 14/2007 and RD 1617/2011. The BioBank protocols are in accordance with the ethical standards of our institution and with the 1964 Helsinki declaration and its later amendments or comparable ethical standards. Formal consent from participants was not obtained because these samples were anonymized.

### Animals

The transgenic murine line TgNN6h was used as a source of brains for preparing PMCA substrate and in bioassay. This line expresses a mutated non-glycosylated human PrP$^C$ at nearly physiological levels (~0.6x) on a murine *Prnp*-null background. This mutated PrP$^C$ contains two asparagine to glutamine substitutions at residues 181 and 197, leading to the elimination of the two N-linked glycosylation sites, in combination with methionine at position 129 (129M). Although it is unable to glycosylate during its post-translational processing, this N181Q/N197Q PrP$^C$ retains its normal intracellular trafficking and membrane location, rendering TgNN6h mice susceptible to prion infection [34].

The transgenic murine line Tg340 was used for bioassay. This line expresses wild-type, fully glycosylated human 129M PrP$^C$ at levels 4-fold those of normal human brain, on a murine *Prnp*-null background [7].

The transgenic murine line BoTg110 was used for bioassay. This line expresses wild-type, fully glycosylated bovine PrPC at levels 8-fold those of normal cattle brain), on a on a murine *Prnp*-null background [4].

## Tissues and inocula

Human brain tissue used for PMCA substrate preparation was obtained from Basque Biobank from Bioef (Berrikuntza + Ikerketa + Osasuna Eusko Fundazioa), Bizkaia, Spain.

Cattle brain tissue used for PMCA substrate preparation was provided by Centro de Encefalopatías y Enfermedades Transmisibles Emergentes, University of Zaragoza, Zaragoza, Spain).

Cattle classical BSE isolate was obtained from the brain of a BSE-affected cow, and provided by Laboratorio Central de Veterinaria, Algete, Madrid, Spain.

Sheep BSE isolate (sBSE) was obtained from the brain of a BSE-affected ARQ/ARQ sheep and was supplied by UMR INRA-ENVT 1225, Interactions Hôtes- Agents Pathogènes, École Nationale Vétérinaire de Toulouse, Toulouse, France.

Pig BSE isolate (pBSE) was prepared from the brain of a terminally ill minipig, inoculated with sBSE (P-1224) [35], and was provided by Centro de Encefalopatías y Enfermedades Transmisibles Emergentes, Facultad de Veterinaria, Universidad de Zaragoza, Zaragoza, Spain.

The vCJD isolate was supplied by Fundación Hospital Alcorcón de Madrid, Madrid, Spain.

All inocula were prepared from brain tissue as 1% (w/v) homogenates in PBS prior to PMCA and bioassay.

## *In vitro* propagation of prions by Protein Misfolding Cyclic Amplification (PMCA)

The *in vitro* prion replication and the PrP$^{Sc}$ detection of amplified samples were performed as described previously [55,58]. Uninfected TgNN6h mice brain homogenates were used as substrates for PMCA. After sacrifice of the animals by $CO_2$ exposure, TgNN6h mouse brains intended for substrate were perfused using PBS + 5 mM EDTA and immediately frozen at -80˚C. The brain substrate (10% brain homogenate) was prepared using a tissue grinder, homogenizing the brain tissue in PMCA buffer (PBS + NaCl 0.15 M + 1% Triton X-100). As controls, 10% normal cow and human brain homogenates were seeded with the same BSE inocula. PMCA reactions were performed by mixing 5 μL of the corresponding inoculum with 45 μL of substrate in 0.2 mL PCR tubes. Each inoculum was assayed in quadruplicate. Tubes were placed on the plate holder of a S-4000 Misonix sonicator (QSonica, Newtown, CT, USA) and subjected to incubation cycles of 30 min at 37˚C without shaking, followed by sonication pulses of 20 s at 80% power. After a 24-h PMCA round, aliquots from the first round were diluted 1:10 in fresh substrate; this procedure was repeated for 15 rounds of PMCA. An equivalent number of unseeded tubes were exposed to the same procedure to control cross-contamination. Ten additional PMCA rounds (i.e. a total of 25 rounds) were performed for each isolate specifically in the TgNN6h substrate, with the aim of generating prions fully adapted to the TgNN6h model while ensuring the absence of any molecule coming from the original inoculum.

For the *in vitro* titration of the original cattle BSE isolate and BSE, sBSE, and pBSE-infected TgNN6h brain homogenates that were used to inoculate BoTg110 mice, 10% brain homogenates were serially diluted ($10^{-1}$ to $10^{-6}$) and used as seeds on a substrate prepared from healthy BoTg110 mice. All the samples were subjected to two serial 24-hour rounds of PMCA in the same conditions described above, and performing 1:10 dilution in between the two serial rounds. PMCA products from the second round were analyzed as described below.

## Mouse bioassay and sample processing

The original prion isolates and the *in vitro*-generated inocula from round 25 of PMCA (termed BSE-PMCA, sBSE-PMCA, pBSE-PMCA and vCJD-PMCA inocula throughout the manuscript) were inoculated in TgNN6h mice, expressing unglycosylated human PrP$^C$. As controls, Tg340 mice, expressing normally glycosylated PrP$^C$, were inoculated with the original isolates. To confirm that the strain properties of BSE prions were maintained upon *in vitro* and *in vivo* adaptation to the non-glycosylated model, BoTg110 mice, expressing wild type bovine PrP$^C$, were challenged with TgNN6h-passaged BSE prions.

In all cases, each mouse received an intracerebral inoculation (20 μL) into the right cerebral hemisphere, using a 50-μL precision syringe and a 25-G needle and under isoflurane anesthesia. The inocula consisted of 1% brain (w/v in PBS) homogenates. To reduce post-inoculation pain, each mouse was given a subcutaneous dose of buprenorphine (0.3 mg/kg). Following inoculation, animals were housed in cages placed in HEPA-filtered ventilated racks and monitored three times per week for onset of neurological signs of prion disease. Mice were culled by cervical dislocation when clinical signs of advanced TSE were detected (i.e. sustained kyphosis, severe ataxia, and poor body condition) or at the end of the study (600–700 days post-inoculation). Brains were removed immediately after euthanasia and divided sagittally. One brain hemisphere was frozen at -80˚C for subsequent biochemical analyses. The other brain hemisphere was placed in 10% formalin fixative for up to 48 h and used for neuropathological studies.

## Biochemical analysis of *in vitro*- and *in vivo*-generated prions

PMCA propagated samples and 10% brain homogenates from prion-inoculated mice were digested using either 85 or 170 μg/mL proteinase K (PK) during 1 h at 42˚C with constant agitation (450 rpm) as previously described [55]. PK concentration was empirically determined for each type of isolate in order to achieve complete digestion in all cases, due to the differences in PrP$^C$ sequence, expression levels and microenvironment. Digestion was stopped by adding loading buffer (4x NuPAGE LDS sample buffer; Invitrogen Life Technologies) and the samples were analyzed by Western blot. Samples were run in parallel with either undigested TgNN6h, human, or cattle substrate, or undigested samples from healthy TgNN6h (unglycosylated human PrP$^C$) or TgWV mice (normally glycosylated human PrP$^C$ [69], as controls. Immunodetection of prion protein was performed with mouse monoclonal antibodies 6H4 (1:10,000) or 3F4 (1:10,000) and visualized using a horseradish peroxidase-conjugated secondary antibody and chemiluminiscence (Super Signal West Pico kit; Thermo Scientific Pierce).

PNGase treatment was performed in 10% brain homogenates from vCJD inoculated Tg340 and TgNN6h animals after PK digestion, using CarboClip PNGase F (Asparia Glycomics) and following instructions from the supplier. Briefly, the denaturing buffer (5% SDS, 400 mM DTT) was added to previously digested brain homogenates at 1:10 dilution and they were incubated at 100˚C for 10 min. After adding the reaction buffer (5 mM sodium phosphate, pH 7.5 with 1% NP-40) and PNGase F at 3 ng/μl, samples were incubated for 1 h at 37˚C. Finally 4x NuPAGE LDS buffer was added and samples analyzed by Western blot as described above.

## Neuropathology

Brains fixed in formalin were embedded in paraffin wax, cut into 4 μm-thick sections and mounted on glass slides. For the evaluation of spongiform lesions, sections were stained with hematoxylin and eosin.

The morphology and brain distribution of PrP$^{Sc}$ deposits were evaluated by immunohistochemical procedures using a protocol described elsewhere [70]. Briefly, after dewaxing and

rehydration, sections were pre-treated with 98% formic acid for 10 min followed by incubation with 4 μg/mL of proteinase K (F. Hoffmann, La Roche) for 15 min. After hydrated autoclaving at 96˚C in citrate buffer for 20 min, immunodetection of PrP$^{Sc}$ was performed by incubating the samples with 3F4 monoclonal antibody (TgNN6h and Tg340 samples, 1:1,000, EMD Millipore, MAB 1562) or 6H4 monoclonal antibody (BoTg110 samples, 1:2,000, Prionics AG) for 30 min at room temperature followed by 30 min of incubation with an anti-mouse Envision polymer (Dako). DAB (diaminobenzidine, Dako) was used as the chromogen substrate.

Amyloid accumulation was also detected in several sections using Congo Red staining, as previously described [71]. Deparaffinized and hydrated brain sections were incubated in an alkaline alcohol-saturated NaCl solution for 20 min following by incubation in a solution of 0.5% Congo Red prepared in an alkaline alcohol-saturated NaCl solution for 20 min. Sections were rinsed through two rapid changes of 100% ethanol and two changes of xylene, and then mounted with DPX.

### Semiquantification of neuropathology results and data analysis

The intensity and distribution of spongiform changes and PrP$^{Sc}$ immunolabeling were blindly evaluated using an optical microscope (Zeiss Axioskop 40), and semiquantitatively scored on a scale of 0 (absence of spongiosis or immunolabeling) to 5 (high intensity of lesion or immunolabeling) in the following brain regions: frontal cortex (Fc), parietal cortex at the level of the thalamus (Pc), septal area (Sa), corpus callosum (Cc), hippocampus (Hc), thalamus (T), hypothalamus (Ht), mesencephalon (Mes), pons (Po), cerebellum (Cbl), and medulla oblongata (Mo).

Lesion profiles and PrP$^{Sc}$ deposits distributions curves were drawn on GraphPad Prism version 6.0 (GraphPad Software, La Jolla, CA, USA).

### Supporting information

**S1 Fig. PNGase digestion of vCJD-infected Tg340 and vCJD-PMCA-TgNN6h mice brains.** Samples were digested with 170 μg/mL PK and 3 ng/μL PNGase F, loaded on SDS-PAGE and subjected to Western Blot using monoclonal antibody 3F4 (1:10,000). Note that PNGase treatment reduced vCJD-infected Tg340 three-band pattern to a single band of approximately 19 kDa, but did not affect the banding pattern of vCJD-PMCA-infected TgNN6h, which in addition retained the ~15-kDa band that we have previously observed in this unglycosylated model, assumed to be an endogenous proteolytic fragment. PNGase digestion did not cause the emergence of the 15-kDa band in Tg340 samples, indicating that it is not the unglycosylated form of a fragment present in Tg340 brains.
(DOCX)

**S2 Fig. Hematoxylin-eosin and Congo Red staining of a florid plaque observed in the thalamus of a sBSE-PMCA inoculated TgNN6h mouse (x40).** The core and the radiating spicules of the florid plaques observed in TgNN6h mice were Congo Red positive, indicating an amyloid fibrils organization.
(DOCX)

**S3 Fig.** Florid plaques observed in the subcallosal area of a Tg340 mouse inoculated with vCJD.
(DOCX)

**S4 Fig. *In vitro* titration of BSE infectivity present in the original BSE inoculum and the PMCA-adapted, then TgNN6h-passaged BSE, sBSE and pBSE inocula.** Two serial PMCA rounds of these inocula were performed on BoTg110 substrate, and the PMCA products were

digested with 85 μg/mL PK, loaded on SDS-PAGE and subjected to Western Blot using monoclonal antibody 6H4 (1:10,000); only the second PMCA round is shown. Note that, while the original inoculum (from cattle) was able to propagate down to a $10^{-5}$ dilution in two rounds of serial PMCA, the TgNN6h-passaged inocula were able to amplify only at a $10^{-1}$ dilution, indicating that their infectious titers were significant lower.
(DOCX)

## Acknowledgments

The authors want to acknowledge the excellent technical assistance of Patricia Piñeiro, Sandra Felices and Daniel Romanos. We also thank MINECO for the Severo Ochoa Excellence Accreditation (SEV-2016-0644). We also want to acknowledge Dr. Zou for the TgWV brain samples provided as control.

We thank the Biobank from Fundación Hospital Alcorcón de Madrid (Madrid—Spain) and Basque Biobank—Fundación Vasca de Innovación de Investigación Sanitaria (Bizkaia–Spain) for providing the human samples.

## Author Contributions

**Conceptualization:** Rosa Bolea, Joaquín Castilla.

**Formal analysis:** Alicia Otero, Tomás Barrio, Hasier Eraña, Juan J. Badiola, Rosa Bolea, Joaquín Castilla.

**Funding acquisition:** Juan J. Badiola, Rosa Bolea.

**Investigation:** Alicia Otero, Tomás Barrio, Hasier Eraña, Marina Betancor, Carlos M. Díaz-Domínguez, Belén Marín, Juan J. Badiola.

**Methodology:** Alicia Otero, Tomás Barrio, Hasier Eraña, Jorge M. Charco, Belén Marín, Rosa Bolea, Joaquín Castilla.

**Resources:** Olivier Andréoletti, Juan M. Torres, Qingzhong Kong, Juan J. Badiola, Joaquín Castilla.

**Supervision:** Juan J. Badiola, Rosa Bolea, Joaquín Castilla.

**Writing – original draft:** Alicia Otero, Tomás Barrio.

**Writing – review & editing:** Alicia Otero, Tomás Barrio, Hasier Eraña, Rosa Bolea, Joaquín Castilla.

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
