## [Decision Letter · Decision Letter 0]

4 Jul 2022

Dear Dr. Castilla,

Thank you very much for submitting your manuscript "Glycans are not necessary to maintain the pathobiological features of bovine spongiform encephalopathy" for consideration at PLOS Pathogens. As with all papers reviewed by the journal, your manuscript was reviewed by members of the editorial board and by several independent reviewers. The reviewers appreciated the attention to an important topic. Based on the reviews, we are likely to accept this manuscript for publication, providing that you modify the manuscript according to the review recommendations.

Sincerely,

Amanda L. Woerman, PhD

Guest Editor

PLOS Pathogens

Neil Mabbott

Section Editor

PLOS Pathogens

Kasturi Haldar

Editor-in-Chief

PLOS Pathogens

orcid.org/0000-0001-5065-158X

Michael Malim

Editor-in-Chief

PLOS Pathogens

orcid.org/0000-0002-7699-2064

Reviewer Comments (if any, and for reference):

Reviewer's Responses to Questions

**Part I - Summary**

Reviewer #1: The normal form of the prion protein PrPC can be variably occupied by glycans at two asparagine amino acids located in the C-terminal core. In the pathological PrPSc form, there are stoichiometric ratios of glycoforms attached to the protein. Whether PrPSc glycosylation is required in maintaining prion strain properties and in facilitating prion transmission within and between species has been an ongoing, fascinating debate for the last 20 years. Any insight on these topics will help further understanding the biology of these agents, including during adaptative events. In this context, Otero and collaborators report on the absence of glycosylation requirements to maintain the strain properties of BSE prions on adaptation to human PrP sequence.

Methodologically, the authors built on transgenic mice expressing unglycosylated human PrP (the previously published tgNN6h mouse line) that was used for bioassay or for brain substrate for in-house PMCA experiments. BSE prions were from cattle or passaged through different farmed species or human under the form of vCJD.

Overall, the study is well constructed, the results are straightforward and manuscript is very clear.

I have several questions and minor comments.

Reviewer #2: Prion protein glycosylation can influence how prions replicate and cause disease in the host in a strain-specific manner. It can also impact cross species formation of prions. The manuscript by Otero et al. examines how glycosylation of the normal human prion protein (PrPC) effects prion formation and disease using BSE prions derived from cattle, sheep, pigs, and humans (i.e. vCJD). They find that the absence of glycosylation in human PrPC does not alter the BSE strain phenotype. They conclude that glycosylation of human PrPC is not essential for preserving either BSE strain properties or the species barrier to transmission of BSE to humans.

This is a straightforward, solid study using a mix of in vitro prion replication assays and in vivo bioassays to assess the role of PrP glycosylation on the transmission and strain properties of BSE. The data are clearly presented and appropriately interpreted and add to our understanding of how prion protein glycosylation can impact prion pathogenesis.

Reviewer #3: Otero et al. describe results that vCJD, BSE & BSE passaged through sheep and porcine can seed formation of prions with an unglycosylated human PrP substrate in PMCA, that subsequently can be passaged back into BoTg110 mice with restoration of BSE phenotype. They also demonstrate biochemical evidence that the protease resistant core and infectivity is maintained with in vitro PMCA propagation to suggest glycosylation is not needed to maintain strain properties with passaging. Strong support to indicate lack of glycosylation does not ultimately impede BSE-initiated conversion of human PrP that maintains overall BSE strain properties is provided with evidence that the generated prions can propagate in BoTg110 mice.

This study provides interesting and further evidence that “strainness” is dictated by the structural backbone of a prion strain, and not fundamentally altered by glycans overall. However, and considering the author’s own data that passaged unglycosylated BSE-seeded prions result in incomplete attack rates in BoTg110 mice, it is this reviewer’s perspective that strong statements such as that glycans do not alter the “strength” of the transmission barrier or are “not relevant” in maintenance of strain properties is not fully indicated by the data here that cannot account for potential subtleties in propagation kinetics or conformation/structures as it relates to glycosylated and unglycosylated forms. Such strong statements should be avoided or rephrased.

**Part II – Major Issues: Key Experiments Required for Acceptance**

Reviewer #1: - In table 1, the authors show that the different BSE/vCJD-PMCA samples readily propagate in tgNN6h mice. Did the authors inoculate tgNN6h mice with the same BSE/vCJD prions amplified by PMCA onto tg340 mice expressing glycosylated PrPC to further examine if the absence of glycans facilitate the propagation of these agents? This is suggested by the observation that vCJD prions propagate at similar attack rate and incubation periods in tgNN6h compared to tg340 mice despite 6-fold decreased PrPC expression levels.

- Did the authors inoculated the BSE/vCJD-PMCA products to tg340 mice to further sustain the view that the strains properties were maintained despite amplification on a deglycosylated PrPC substrate?

- The authors found that PrPSc found in the brains of tgNNH6 mice inoculated with BSE/vCJD-PMCA samples exhibit a vCJD-like electrophoretic signature, with unglycosylated PrPSc migrating at 19-20 kDa. They also found an additional band at 15 kDa. Is this band also observed in tg340 mice inoculated with vCJD/BSE samples after PNGase deglycosylation?

- It is a bit surprising that BSE/vCJD-PMCA passaged twice on tgNN6h reinfected parental bovine PrP mice at incomplete attack rate and longer incubation time than cattle BSE. As the PrPSc signature found in the brain of the sick mice is similar to that of BSE in the same model, the authors conclude that the strain properties of BSE have been recovered. Strictly speaking, a second passage would be necessary for a more definitive conclusion. To explain this apparent discrepancy, the authors suggest that the infectivity levels of BSE/vCJD prions in tgNN6h mice may be lowered, due to the low levels of PrPC expression in these mice. At least two articles contradict this hypothesis: 1) heterozygous mice harbour similar infectivity levels than wild-type mice at terminal stage, despite extended incubation time (Bueler, et al., 1994); 2) scrapie prions infectivity levels are similar between wt animals and overexpressing mice (Douet, et al., 2014). To address this issue, the authors could titrate the seeding activity of their brains compared to cattle BSE brain.

Maybe BSE/vCJD serially passaged on tg340 mice would have been a better control for back-passage into bovine PrP mice.

Providing more information or data would strengthen the view that unglycosylated BSE prions did not evolve phenotypically.

- Histopathological analyses unambiguously show the presence of florid plaques in the inoculated tgNNh6 mice. This type of plaque is archetypical of vCJD/BSE propagated on the human PrP sequence. Does this suggest that in mice expressing wild-type human PrP or even in humans, these plaques are preferentially formed from unglycosylated PrPSc? The authors may have referred to the work published by Sevillano et al. who showed that glycans reduced plaque formation in prion disease (Sevillano, et al., 2020).

References

Bueler H, Raeber A, Sailer A, Fischer M, Aguzzi A, Weissmann C (1994) High prion and PrPSc levels but delayed onset of disease in scrapie-inoculated mice heterozygous for a disrupted PrP gene. Mol Med 1:19-30

Douet JY, Lacroux C, Corbiere F, Litaise C, Simmons H, Lugan S, Costes P, Cassard H, Weisbecker JL, Schelcher F, Andreoletti O (2014) PrP expression level and sensitivity to prion infection. J Virol 88:5870-5872

Padilla D, Beringue V, Espinosa JC, Andreoletti O, Jaumain E, Reine F, Herzog L, Gutierrez-Adan A, Pintado B, Laude H, Torres JM (2011) Sheep and goat BSE propagate more efficiently than cattle BSE in human PrP transgenic mice. PLoS Pathog 7:e1001319

Sevillano AM, Aguilar-Calvo P, Kurt TD, Lawrence JA, Soldau K, Nam TH, Schumann T, Pizzo DP, Nystrom S, Choudhury B, Altmeppen H, Esko JD, Glatzel M, Nilsson KPR, Sigurdson CJ (2020) Prion protein glycans reduce intracerebral fibril formation and spongiosis in prion disease. J Clin Invest 130:1350-1362

Reviewer #2: None

Reviewer #3: Additional comments:

1) Does a 1% compared to 10% brain homogenate (10-fold difference) sufficiently reconcile the discrepancy between the results reported here in BoTg110 mice, when compared to those previously reported? Authors should discuss titers, and if a 10-fold decrease based solely on titer would be expected to result in lack of disease in these models, or if this is speculative.

2) The authors reason that the incomplete attack rate above is likely attributable to 1) a 0.6x expression level of the TgNN6h mice that ultimately results in less PrPSc accumulation for subsequent passage and 2) Use of 10-fold less material (see point 1). However, these statements rely on the assumption that less PrPSc eventually accumulates that, per amount brain tissue, would be available for subsequent passage. Evidence should be provided/discussed that less PrPSc is known to ultimately accumulate in the TgNN6h mice when compared to WT.

3) Why were different PK conditions used with TgNN6h propagated materials (Fig 2)? If different PK conditions were required for efficient digestion, does this argue against complete maintenance of all strain properties? For direct comparison, the same PK concentrations should be used. The resolution to a single band for the TgNN6h substrate is interpreted to indicate that BSE, sBSE, pBSE and vCJD yield the same product. However, sBSE and vCJD seeded PMCA prions appear less protease resistant (less material following digestion on blot) which could indicate differences in assembly and/or conformation. Digestion at different PK amounts, including a limited PK digestion, or another approach to assess conformation of the BSE, sBSE, pBSE, and vCJD PMCA products would better indicate if conformational differences, albeit ultimately not prohibitive towards propagation in the animal models, do exist and contribute to differences in PK resistance observed.

**Part III – Minor Issues: Editorial and Data Presentation Modifications**

Reviewer #1: - Lane 176 : the authors mention that the disease incidence in tg340 mice inoculated with BSE/vCJD isolates differ from that published by Padilla et al. (Padilla, et al., 2011). They pointed out correctly that inoculations were made at 1% in their lab compared to 10% in Padilla’s work. My understanding is also that the isolates used (cattle BSE, sheep BSE, vCJD) are different and may harbour different levels of infectivity.

- Lane 255: I would replace “…cross-species transmission of prions to humans…” by “…the potential cross-species transmission of prions to humans…” to avoid any misinterpretation.

- Lane 357: 127S and not 126S

Reviewer #2: Minor comments:

1) The authors should clarify the statement on line 76 that “expression of PrPC in the cellular membrane is necessary for the development pf prion disease”. Prion disease does in fact develop in mice that express PrPC without the GPI anchor (PLoS Pathog 6(3): e1000800. (2010)), which is secreted from the cell and is not anchored to the cell surface.

2) In Figure 3, PrPC substrate from TgZW mice is shown on the western blot. TgZW mice are only mentioned in the Figure 3 legend and are described as expressing “normally glycosylated human PrPC”. More information on this line needs to be provided in the Methods if they are going to be used as a control. Why wasn’t substrate from Tg340 mice, which also make normally glycosylated human PrPC, used on these blots? Since those were the mice used for the in vivo bioassay, that would seem to be a more appropriate control.

3) There is no reference to Figure 5 in the text where the data are discussed (lines 242-245) and this should be added. In addition, the red arrows in Figure 5 need to be defined in the legend. Presumably, they indicate passage history, but this is not clear.

4) The description of the neuropathology in the TgBo110 mice is quite brief (lines 246-248) and makes no mention of the fact that it is significantly different from that in the TgNN6h mice. PrPSc deposition is fine-punctate and plaque-like in the TgBo110 mice (Figure 6) versus florid plaques in Tgnn6h mice (Figure 4). In addition, the pattern of spongiform change is clearly different (compare top panels in Figures 6A and 4A). The neuropathology of BSE is generally similar when transmitted across multiple species so the difference in pathology between the two transgenic mouse lines is somewhat unexpected. It would be helpful if the authors discussed why they see these differences in a bit more detail, especially with regard to what the data imply about whether PrPC glycosylation or PrPSc glycosylation influences the phenotype of BSE PrPSc deposition in vivo.

Reviewer #3: Typo on line 387 “on a on a”

PLOS authors have the option to publish the peer review history of their article (what does this mean?). If published, this will include your full peer review and any attached files.

Reviewer #1: No

Reviewer #2: No

Reviewer #3: No

Figure Files:

Data Requirements:

Reproducibility:

References:

---

## [Editor Report · Decision Letter 1]

27 Sep 2022

Dear Dr. Castilla,

We are pleased to inform you that your manuscript 'Glycans are not necessary to maintain the pathobiological features of bovine spongiform encephalopathy' has been provisionally accepted for publication in PLOS Pathogens.

Best regards,

Amanda L. Woerman

Associate Editor

PLOS Pathogens

Neil Mabbott

Section Editor

PLOS Pathogens

Kasturi Haldar

Editor-in-Chief

PLOS Pathogens

orcid.org/0000-0001-5065-158X

Michael Malim

Editor-in-Chief

PLOS Pathogens

orcid.org/0000-0002-7699-2064
---

## [Editor Report · Acceptance letter]

4 Oct 2022

Dear Dr. Castilla,

We are delighted to inform you that your manuscript, "Glycans are not necessary to maintain the pathobiological features of bovine spongiform encephalopathy," has been formally accepted for publication in PLOS Pathogens.

Best regards,

Kasturi Haldar

Editor-in-Chief

PLOS Pathogens

orcid.org/0000-0001-5065-158X

Michael Malim

Editor-in-Chief

PLOS Pathogens

orcid.org/0000-0002-7699-2064